# Barriers and Facilitators to Healthy Eating for Shift-Work-Registered Nurses in Hong Kong Public Hospitals: An Exploratory Multi-Method Study

**DOI:** 10.3390/nu17071162

**Published:** 2025-03-27

**Authors:** Pui-Lam Ling, Zhi-Yang Lai, Hui-Lin Cheng, Ka-Hei Lo

**Affiliations:** 1School of Nursing, The Hong Kong Polytechnic University, Hong Kong SAR, China; puilam.ling@connect.polyu.hk; 2Department of Food Science and Nutrition, The Hong Kong Polytechnic University, Hong Kong SAR, China; zhiyang.lai@connect.polyu.hk; 3Research Institute for Smart Ageing, The Hong Kong Polytechnic University, Hong Kong SAR, China

**Keywords:** barriers, eating behaviors, facilitator, nutrition, registered nurses, shift work, workplace

## Abstract

**Background/Objectives:** Shift work has profound effects on the health and dietary habits of registered nurses, especially in Hong Kong, where cultural and systematic barriers can pose a challenge. This study investigated the dietary habits of shift-working nurses in public hospitals, identifying barriers and facilitators to healthy eating using a mixed-methods approach. **Methods:** Nine subjects (five females and four males: mean = 35.6, SD = 8.4 yrs) filled out a validated food frequency questionnaire and a 3-day dietary record followed by photovoice and semi-structured interview. **Results:** The findings indicated that all participants reported insufficient dietary fiber intake and eight out of nine exceeded sodium intake recommendations. The key barriers included emotional eating triggered by work stress, inconsistent schedules, limited availability of nutritious foods, and workplace social dynamics. The facilitators included workplace support, positive peer influence, and family involvement in meal planning. **Conclusions:** This paper focuses on the necessity for health care institutions to create and implement nutritional instructions specific to shift workers, maintain appropriate meal breaks, and build a positive work environment. These interventions may be used to enhance nurses’ eating habits and well-being.

## 1. Introduction

Nursing is the largest health care profession globally, with an estimated 29 million nurses worldwide [1]. In Hong Kong, more than 75% of nurses are licensed as registered nurses (RNs) [2], with approximately 65% of them working in public hospitals [3]. To provide 24-hour continuous care to patients, the working nature of nurses in public hospitals typically involves rotating shifts, including morning, afternoon, and overnight duties [4]. While shift work effectively maintains the continuity of services, it adversely disrupts nurses’ health in a negative way [5]. In general, shift work affects nurses in different aspects such as occupational stress, sleep deprivation, gastrointestinal symptoms, and cardiovascular diseases [6,7]. Nurses with frequent night shifts (i.e., more than three times per week) are prone to overweight or obesity due to overeating with unhealthy dietary choices at night [8].

Healthy eating is defined as having a balance of different foods and nutrients in the diet for maintaining health and well-being [9]. Maintaining a good and balanced dietary pattern could help to promote health and prevent non-communicable diseases, such as hypertension, hypercholesterolemia, overweight etc. [10]. Various dietary guidelines have been provided to promote healthy eating for nurses in different countries or regions. For instance, the Centers for Disease Control and Prevention (2020) in the United States advised shift work nurses to avoid sugar-rich products and low-fiber carbohydrate foods [11]; In Hong Kong, the Dietitians Association (2023) suggested that individuals should have at least 2–3 servings of fruits and vegetables [12]. Nevertheless, nurses did not follow the existing guidelines and still engaged in unhealthy eating habits [6]. Han et al. (2016) documented that nearly 65% of nurses in Korea tended to consume less protein, vegetables, and fruits during work due to a lack of time for preparing meals [13], while nurses in Canada and the United States consumed high-fat and calorie snacks such as chips and fried food during their night duties to stay awake during the long shift [5]. According to a large cross-sectional survey conducted by Wong et al. (2010) which examine the associations between shift duties and unhealthy eating habits among 378 Hong Kong nurses, over 70% of the respondents reported having the habit of overeating due to stress and fatigue [4]. This unhealthy eating behavior involved the intake of high-sugar, canned, and processed food to overcome cravings during work, especially for nurses working night shifts four times per month.

Despite the well-documented positive relationship between shift work and poor dietary habits among nurses, research specifically exploring the underlying facilitators and barriers associated with healthy eating among nurses is limited [5]. Furthermore, the literature from western countries might not be suitable to apply to the Chinese population due to different working hours, the nurse-to-patient ratio, and the eating culture [5,14]. Furthermore, existing studies heavily relied on either quantitative or qualitative methods, which may not provide holistic insights into current nurses’ dietary choices and the factors affecting dietary habits during work. With no study having been published in China before, the aim of this study was therefore twofold: first, to understand the current dietary behaviors of Hong Kong shift work nurses working in public hospitals, and second, to explore the barriers and facilitators to healthy eating during work through the combination of qualitative and quantitative methods.

## 2. Materials and Methods

### 2.1. Study Design

This study was designed with an exploratory multi-method approach, in which the first step was to collect quantitative data to understand the current dietary behaviors of nurses by completing the food frequency questionnaire (FFQ), and then to collect qualitative data by filling the 3-day dietary food record and conducting a semi-constructed interview to obtain a thorough picture on the factors that affected healthy eating during work. For the qualitative data, the photovoice research approach was adopted, which has been used in nutritional and dietetic research [15]. By utilizing photographs taken that conveyed factors affecting healthy eating, participants could present their own perspectives and experiences in healthy eating, and acknowledge and explore the reasons behind their daily eating habits [16].

### 2.2. Settings and Participants

To be eligible for this study, the participants had to be full time RNs working with shift work in Hong Kong public hospitals, and they were required to have at least 6 months of night shift work experience with at least 4 overnight shifts per month [4]. Additionally, they could take photographs and share their experiences. Participants were excluded if they worked in private hospitals or other allied health settings, such as nursing homes. RNs who were pregnant or suffering from chronic diseases were also excluded. Snowball and purposive sampling were adopted during the recruitment period by circulating a recruitment poster via social media platforms (Facebook and Instagram) and emails addresses obtained from the PI’s professional network. Interested parties were directly contacted by the PI through email or phone call and the eligible participants were chosen according to the inclusion criteria of the study. In the common photovoice qualitative methodology, it is recommended that the sample size is 6–10 participants to avoid data saturation [17]. As the last two interviews did not reveal any information that was new to the study, the recruitment was concluded after interviewing 9 nurses.

### 2.3. Data Collection Methods

The following study tools were used to examine the quantitative and qualitative data affecting nurses’ maintenance of a healthy diet during work.

#### 2.3.1. Sociodemographic Data

The characteristics of the population were reported with sex, age, marital status, working department, year of working experience, number of night shifts per month, and Body Mass Index (BMI). BMI was used to classified whether the participants were underweight, normal, overweight, or obese [18].

#### 2.3.2. Food Frequency Questionnaire (FFQ)

A 60-item FFQ was used to examine the frequency of consumption of specific foods and beverages over the past one month (please see Appendix A) [19]. This was a validated dietary assessment tool widely used in Hong Kong to understand individuals’ eating patterns and to estimate any prevalence of deficient or excessive nutrient intake [19]. The levels of seven common macro- and micro-nutrients (carbohydrate, sugar, total fat, saturated fat, sodium, protein, and total dietary fiber) were calculated, and the fulfillment of dietary recommendations was referred to the Chinese Dietary Reference Intakes (DRIs) [20]. The FFQ had good reproducibility (r ≥ 0.75) and relative validity (r ≥ 0.70) in assessing the dietary intake of phosphorus, water, grains, wine, protein, carbohydrates, total fat, magnesium, and eggs [19].

#### 2.3.3. Three-Day Dietary Record

This was a precise, detailed food and beverage intake description record to indicate what, when, where, and the quantity of food and drinks that an individual had taken over the day [21]. It could help to reveal information that might not be observed in the FFQ, such as the rationales causing a deficiency or excessive intake of specific nutrients by checking their eating patterns in specific food or beverage groups [22]. Participants were required to record for 3 working days within a 2-week period, covering at least one night shift.

#### 2.3.4. Photovoice

To supplement the observations from the FFQ and 3-day food records, participants were asked to photograph anything (e.g., food, eating habits at work, working environment) with their smartphones that could convey the challenges and opportunities to healthy eating per the recorded working day (please see Appendix A). A minimum of 2 photos with non-blurry, clear captures were required in each food record day, having a total of a minimum of 6 photos taken for the whole study [17]. The use of photos could help to strengthen the dietary records and facilitate the recall of dietary intake during the interview. All these could help to generate complementary data sources and verify the research findings, and thus increase their validity [23]. No photos of human objects were allowed unless consent was provided to ensure individuals’ privacy and confidentiality [24].

#### 2.3.5. Semi-Constructed Interview

After collecting all photos, a face-to-face semi-constructed interview according to the photos taken was conducted. The interviews were guided with the “SHOWed” methodology in a photovoice study, starting the dialog with the following questions [15]: (1) What does the photo show?; (2) What is really happening here?; (3) How does this relate to your lives?; (4) Why does this situation exist? and (5) What can be done about it? Additional questions regarding their description and their reasoning concerning food choice and eating habits during shifts, plus enablers or inhibitors to healthy eating during work, were asked according to the flow of the interview.

### 2.4. Procedures

The data collection was carried out from March 2024 to July 2024. It was divided into 3 phases:

Phase 1—meeting prior to data collection: participants were given an information sheet and their signed, informed consent was obtained; the information sheet was explained in person by the first author before the commencement of the study. They were also required to complete the demographic questionnaire and the FFQ to obtain the quantitative data. The photo-taking activity and 3-day dietary record were then introduced to the participants with the demonstrations following instructions. Participants were also given the chance to perform a return demonstration on photo taking and completing the food record. The training did not include any guide to or definition of “healthy eating”, which avoided influencing the participants’ perception when they performed the photo taking [25].

Phase 2—photo-taking activity with 3-day dietary food record: after receiving training on the study, the participants were asked to finish the dietary records with photo taking in any 3 shift workdays within a 2-week period (starting from the date of training) [26]. It was suggested that they fill in the food record once they had consumed it so as to prevent omission of food items [21]. In each recorded day, at least 2 photos were supplemented to reflect the factors influencing healthy eating during work and sent to the first author password-secured [17].

Phase 3—a face-to-face semi-constructed interview with photos taken and 3-day dietary records: the face-to-face individual semi-constructed interviews guided with the “SHOWed” methodology were carried out in the 3rd week after photo collection to have an in-depth discussion about their experiences and reflections on any barriers and facilitators towards healthy eating during work. The interviews were conducted in the quiet classrooms of the Hong Kong Polytechnic University and recorded with audio recorders with the participants’ consent. Non-verbal cues such as facial expression and pauses were noted down on the transcripts.

### 2.5. Data Analysis

The quantitative data of the FFQ and 3-day dietary records as well as demographic data were analyzed with descriptive analysis through SPSS 28.0 [27]. The data from the FFQ were computed using the ESHA Research Food Processor SQL: Nutrition Analysis and Fitness Program (Copyright 2022, ESHA Research) [19]. The intake levels of nutrients were then compared to the Chinese Dietary Reference Intakes (DRIs) established by the Chinese Nutrition Society [19].

Following the tradition of Elo and Kyngäs (2008)’s [28] steps of content analysis, the qualitative data collected from the semi-constructed interviews were transcribed and translated verbatim from the audio recordings and merged with the notes taken during the interview. The data were then rechecked with the original recordings and sent back to participants to check that the statements represented their meaning as accurately as possible. After verification with the participants, the contents were coded, identified, grouped and categorized into sub-categories and then main categories for reporting [28]. Data interpretation was also checked regularly with the research team to ensure trustworthiness. To further ensure confidentiality, the anonymity of the data was ensured with participant numbering [29].

### 2.6. Ethical Considerations

This study had ethics approval from the Departmental Research Committee, Hong Kong Polytechnic University (Reference Number: HSEARS20240125004) to ensure participants’ confidentiality, informed consent, and right to withdraw from the study. Photographs and electronic files were stored on a secured and encrypted password-protected computer. All of the hard copies, audio tapes, and electronic data would be destroyed by mechanical shredding 5 years after the completion of the study.

## 3. Results

### 3.1. Demographic Characteristics

All nine participants who enrolled participated with no dropouts. There were five females and four males working in various working departments in Hong Kong public hospitals such as medical, surgical, intensive care units, etc. (Table 1). The working experience of the participants ranged from 3 to 25 years. For the BMI classification, five of them were in normal range, one was underweight, two were overweight, and one was obese.

### 3.2. Prevalence of Excessive or Deficient Dietary Intake

The intakes of carbohydrate, sugar, total fat, saturated fat, sodium, protein, and total dietary fiber were computed for analysis. To determine whether the participants were eating healthily or not, the results from the FFQ were compared with reference to the DRIs [19,20]. Carbohydrate, total fat, and saturated fat were calculated according to the percentage of daily energy distribution (Figure 1), while sugar, sodium, protein, and total dietary fiber were calculated using the unit of milligram (mg) or gram (g) for comparison (Figure 2).

Eight out of nine of the respondents had excessive sodium and total fat intake, while 100% of the respondents had a deficiency in carbohydrate and total dietary fiber intake.

### 3.3. Barriers and Facilitators to Healthy Eating During Work

The nine individual semi-constructed interviews with photovoice methodology averaged 73 min in length, with a range of 60 to 90 min. Four categories of individual, social, organizational, and environmental factors emerged from the data to reveal barriers to and facilitators of healthy eating, with each category having subcategories (Table 2).

#### 3.3.1. Barriers to Healthy Eating

##### Individual Factors

Comfort food and emotional eating

Most participants reported that food with a high carbohydrate and sugar content such as ice cream and cola could give them a sense of contentment and increased energy levels after intake. They perceived the enjoyment of food more than its nutritional value, as long as the food could relieve their emotions, especially during a bad day or when experiencing hardship:


*“It was a stressed and busy day to me. Why would I still need to eat badly in a worse day? It’s not the time to mention eating healthy, I would rather go for the food I love.”*
(P5, operating theater, normal BMI)

Participants also described that they would like to consume excessive food in response to negative emotions, but not necessarily because of hunger. This usually happened after a long working day or during a stressful situation. They did agree that only eating could help to resolve stress by gaining a sense of satisfaction and relaxation; nevertheless, the amount, frequency, and the types of comfort food taken were not in consideration. Participants could easily exceed their required nutrition amount per day, and they sometimes might overeat:


*“I love eating Siu Mai. Especially when it was on promotion, I can eat 20 Siu Mai at once after work! Sometimes I would also drink cola as well, make it a full meal!”*
(P3, OBS, obese) (Appendix A)


*“I felt so stressed and nervous during work, which made me double the eating portion after work when I needed to be the nurse in-charge.”*
(P7, medical, overweight)

Another overweight ICU nurse added


*“That night shift was incredibly busy, and the cases were really complicated…I went to 7–11 and bought the (12 bottles) Coca-Cola. The soda pop and carbonated gas kind of giving me a sense of relief after drinking and belched it out. Also, cola is sweet enough to replenish sugar level and energy. Drinking water could not achieve this effect at all. This (the photo) is only for 4 persons to consume, each of us need to relax in the long night shift…”*
(P8, ICU, overweight) (Appendix A)

b.Palatability

The dynamic and subjective perceived taste intensity could affect one’s food choices and intake. Some of the participants perceived that food from the cafeterias in the hospital was always oily and not tasty enough, and would like instead to go for food deliveries. One of them even perceived that “Healthy food are never tasty”, and she refused to have any healthy food intake:


*“Healthy food are never tasty, Tasty food are never healthy. No one will eat healthy food at all! At least I am one of them.”*
(P7, medical, overweight)

c.Personal perception, attitudes, and health issues

Individuals’ perception towards healthy eating also affected by one’s dietary behaviors. One obese nurse perceived herself failing to lose weight and maintain a healthy diet as she loved eating: *“No need to try losing weight as it is impossible to do so, even I have sought help from the dietitian but I always struggle with food and weight. I love eating. Eating is my life! Especially in the hardship and busy work. As I have said I love Siu Mai, I cannot live without it.”* (P3, OBS, obese)

Some male participants having the concept that “men barely cook”, all the meal planning was the responsibility of the women, or of the delivery service. This social norm made them become less active in choosing food and if the man’s wife or family cooked with unhealthy food, he would definitely just follow and maintain an unhealthy diet: *“I don’t know how to cook, but I think these things are better for women than men. What my wife cooked then I will eat. My meal planning is handled by her.”* (P9, surgical, normal BMI)

Another concern with health problems also affected nurses in maintaining healthy eating. One underweight Accident and Emergency department nurse claimed that she got gastric reflux and stomachache when she was on night shifts or was in charge. This affected her appetite and she may need medication to resolve the problems.

##### Social Factors

Colleagues’ influence/appreciations from patients

“Food sharing” was quite common among the nurses during work. In order to fit into social groups, participants usually followed the majority in food options for food delivery during work, usually 4–5 times per week. Some colleagues would order what they wanted to and never considered any nutritional values behind their choices:


*“…whatever they (colleagues) ordered, I would just follow and share the food, regardless its nutritional value. Share dishes is happier compared to eating alone…during night shift, we would brought back many night-food with high carbohydrates such as dim-sums, fried food, or other sweet soups as we knew that we had to stay up all night. Therefore, we brought back a lot of food to share.”*
(P4, pediatric ICU, normal BMI)

On the other hand, colleagues bringing back souvenirs from trips (usually snacks) became a norm in most of the wards.


*“Colleagues would usually bring souvenirs back when they ended their annual leaves. Usually on each Mondays the table will be full of snacks, such as potato chips, chocolate. You cannot escape or avoid it even you just ate one or two bites whenever you enter the pantry to have a rest.”*
(P1, medical, normal BMI)

It also happened when patients or patients’ relatives brought gifts for the health care professionals as a reward, especially in wards with a high patient turnover rate such as the surgical ward, operating theatre, etc. The continuous supply of these tasty bites undoubtedly affected nurses’ healthy eating and built up the frequent snacking behavior of high-sugar or calorie snacks during work:


*“It’s usually 5–6 times a month (patients bringing gifts to the ward) … Once have a time that two patients bringing assorted cakes at the same time, making everyone getting a cake as meals!”*
(P9, surgical, normal BMI) (Appendix A)

b.Family influence

Some participants brought their own meal from home, and followed the eating habits among their family members. Nevertheless, their families insisted on cooking with fried, extreme taste, or spicy dishes, as well as using lots of oil during cooking. Dining out at night was also common in most of the families, and the frequent intake of high-salt and oily food might affect nurses’ health as well:


*“My parents usually off work late, sometimes we did not cook at home and dine out in late night, like 9 or 10 p.m. I often having some snacks to resolve my hunger while waiting them back home. Sometimes I might just bring the left-over when we dine out, which was oily and high salt level.”*
(P8, ICU, overweight)

##### Organizational Factors

Shift work nature

Participants described that it was inevitable and compulsory for them to be on shift up to 6 days per week and to work with long working hours, up to 12 h. Due to the heavy and stressful workload in shifts with a high nurse to patient ratio (~1: 12–15), most participants experienced a significant disruption to their circadian rhythm. For instance, they could not adjust between day and night, and might need to eat a big meal after the night shift when they woke up after the long day.


*“The rotating shift really kills me with 6 working days and one day off per week. I worked 6 night shifts last month, not to mention the long working hour of 10–12 h per day. The workload is really heavy, I am really ready to submit the resignation form if the situation continues…”*
(P1, medical, normal BMI)

Furthermore, some participants mentioned that there were only two nurses looking after 40–50 patients during their night shift routines. They needed more energy to stay awake and therefore prepared something heavy with high carbohydrate.


*“A shift usually works at 7 a.m., and lunch at 1:30 p.m. I can only have my so-called full meal at 2 p.m. until I finished my handover. It is very hungry for me and I usually grab any food from the pantry whatever is it, until getting full.”*
(P6, surgical, normal BMI)

b.Sudden change of work schedule

Some participants mentioned that unscheduled arrangements were common and frequent in their wards. Some colleagues might need to replace their colleagues on sick leave ad hoc at very short notice (e.g., a few hours before work). From their perspective, it was very difficult for them to adjust their diet promptly, not to mention eating healthy. One of the medical nurses even had ice cream and chips after that “ad hoc” working day to replenish the busy day.


*“There was no use to prepare meals or snacks for the next day, as you will never know if you need to work or not. (The ward manager) giving you an off suddenly, begging you to work tomorrow…we (the colleagues) were all used to it.”*
(P7, medical, overweight) (Appendix A)

c.Meal break availability

Most of the participants reported that the meal time in each shift was usually one hour. However, their meal time was frequently disrupted due to patients’ changed conditions and sudden incidents. They commented that nurses would never bring decent meals or eat slowly during the meal time, they would bring handy snacks and convenient food instead. Meal times even became a stress for them as they did not know when to pop out when there is a need to:


*“The “beep beep” sound from the medical devices scared me. Whenever it was ringing, every nurse would get full attention to the patient and stop doing anything, not to mention finishing the whole meal. By the time I back to the pantry, I did not have the appetite to eat the “leftover” food anymore.”*
(P4, pediatric ICU, normal BMI) (Appendix A)

##### Environmental Factors

Cost and affordability of fast food

When being probed with the reasons for choosing unhealthy foods, most participants explained that expensive healthy food and the low cost of unhealthy fast food rendered them more prone to choose unhealthy food (e.g., junk food, snacks) during work. A participant shared that prices were always taken into his consideration when choosing food: *“Although it is convenient and sometimes good to have healthy soup to drink during work, I would still take the price as consideration (HKD 50–70 per pack)…”* (P7, medical, overweight). Another nurse from the operating theater commented that the cost difference between dining out and preparing the same dish at home (i.e., salad) was around HKD 20–30, which was a huge price difference for her. She would rather prepare the meal by herself or choose some convenient food with a lower price during work.

b.Availability and accessibility of healthy food during work

Some participants expressed that fast food was easily accessed across the hospital, such as in vending machines and mobile food stores. It was easy for them grab rice balls, soft drinks, biscuits, etc. when they were in a hurry. A few participants explained that because the cafeterias in the hospitals provided limited food choices and usually high-calorie food and drinks, they had no choice and were forced to order unhealthy food from the menu:


*“The dishes (from the cafeteria) were always added with MSG…a free soft drink would be included when you order meals. It certainly encouraged nurses to have the unhealthy drink per day.”*
(P2, A&E, underweight)

#### 3.3.2. Facilitators to Healthy Eating

##### Individual Factors

Palatability/perceived healthiness in food choice

Some nurses were entirely good with vegetables, fruits, grains, or other food with high nutritional value. They perceived that fruits and vegetables could help to enhance immunity, making them feeling more refreshed and healthier.


*“I do like having healthy food, it can make the condition of my skin better as well. Fruits after meal is a must to keep me healthy.”*
(P4, pediatric ICU, normal BMI)


*“…I usually eat fruits with high vitamin C which helps to enhance immunity, such as orange, golden kiwi. Especially when I was moved out from my family, I paid more attention to the food I eat and consider to eat healthier”*
(P6, surgical, normal BMI)

b.Personal knowledge

One of the participants had studied the Post-registration Certificate Course (PRCC) on diabetes mellitus (DM) and understood the foods to avoid or quantities that should be taken to prevent herself having a risk of DM. When educating DM patients on healthy food choices, this nurse could also benefit from it by gaining adequate knowledge and became a role model for patients in maintaining a healthy diet:


*“…If I really want to eat starch, I will choose to eat buckwheat noodles. Sometimes I would choose to eat macaroni, which is the lower Glycemic Index food.”*
(P1, medical, normal BMI)

##### Social Factors

Colleagues’ influence

Participants described the positive influence from colleagues in different ways. For example, some participants mentioned that to maintain hydration during work, some of their colleagues brought large water bottles and some of the participants followed the habit and drank more water or tea together (Appendix A). Additionally, some colleagues also considered healthy food options when ordering delivery during meals: *“if we order rice noodle as lunch, we would change the noodle base to vegetable, and to avoid eating spicy soup. Sometimes we did not choose the delivery with fried food, oily dishes. All of my colleagues would love to eat healthier during work.”* (P1, medical, normal BMI)

b.Family influence

According to some participants, a family’s cooking style also shaped the positive side in maintaining healthy eating. They mentioned that their parents or mother-in-law would cook for them occasionally with healthy and balanced dishes such as fish soup or vegetables as a retreat after a long working day (Appendix A).

For families with children, they needed to consider the food options wisely as it would affect child’s development. For example, children were in a nurturing state and more protein and fibers should be taken; sugar or junk food should be avoided. This also helped the parents to shape a healthy diet in their daily life and build up a healthy eating habit diet at home and during work:


*“I would tell her (maid) that every meal must include meat and vegetables, sometimes fish, and more protein. As I have 2 kids, they need to have a balanced diet and the food choices of the children must be prioritized, especially they are in a developmental period. We did not let them consume fried food or ice cream, so as to us, we have to be a good role models for the children.”*
(P4, pediatric ICU, normal BMI)

##### Organizational Factors

Workplace support

A few participants mentioned that their working units were united and maintained a healthy eating environment in the working department. The ward manager and seniors would join the food gathering and promote healthy eating in the ward by food sharing and adding different cooking utensils in the pantry such as a microwave, oven, which allowed nurses to cook with healthy foods during work and promoted a positive atmosphere.


*“(talking about her colleagues were united in eating healthy at work)… Even my senior colleagues have begun to eat healthily, but I am not sure if it was because they saw all the young nurses eating healthily, so they would like to join us during meal time. My boss even supported us by adding on some cooking utensils such as microwave oven in the pantry and encourages staff to make their own food during work.”*
(P1, medical, normal BMI)

Some hospitals would hold a fruit day during the year to promote healthy eating in the hospital. All of these become motivations for colleagues to maintain healthy eating during work.


*““Fruit day” is held every year in my hospital, which aims to encourage staff to maintain healthy diet. Each of the staff will be given a kind of free fruit, such as banana, orange, apple during meal time. However, we might not always have time to enjoy the fruit during the day, so sometimes I will bring the fruits back home and take it afterwards. I do agree holding this kind of activity could help staff to maintain healthy eating during work, however, the frequency should be increased, maybe holding quarterly to maintain the sustainability.”*
(P2, A&E, underweight)

##### Environmental Factors

Cost and affordability of healthy food

Some nurses claimed that a delivery for lunch or dinner was expensive and not healthy at all, they considered cooking at home by themselves with less oil and salt, making a healthy meal during work. It usually happened when nurses moved out from the family and needed to be responsible for all their expenses.

*“Sometimes I enjoyed cooking with healthy food. It was not worthy to dine out with over-priced dishes with simple and cheap ingredients.”* (P6, medical, normal BMI)

b.Availability and accessibility of healthy food during work

A few participants said that some cafeterias in the hospitals had the nutritional labeling printed on the menu, showing the number of calories per dish and allowing nurses to choose wisely when they ordered meals. They described that some even had a vegan-friendly menu which catered to the needs of vegetarians and nurses who were interested in veganism as well:


*“I am not a vegan but sometimes I still wanted to have some meat-free meals… It tasted so-so, but that was also a choice for us.”*
(P4, pediatric ICU, normal) (Appendix A)

## 4. Discussion

The current study discussed both barriers to and facilitators in maintaining healthy eating during work in individual, social, environmental, and organizational aspects. It presented evidence that social and organizational factors affecting healthy eating among nurses fall on both the positive and negative sides, which is in contrast to those reporting only on one side in the literature [30]. The qualitative and quantitative data complemented each other and reflected a holistic picture of healthy eating among the nurses.

### 4.1. Overall Deficiencies in Dietary Fiber and Excessive Sodium Levels

All of the participants had an excessive sodium intake and inadequate dietary fiber according to the results of the FFQ. According to the Population Health Survey undertaken by the Center for Health Protection in Hong Kong in 2022 [31,32], approximately 78% of the respondents aged 25–64 consumed vegetables and fruits once per day, with 98% of the respondents not having at least five servings or 400 g of vegetables per day. On the other hand, 77.2% of the participants claimed to dine out once per day, with a mean salt intake of 7.6 g. The report proved that the proportion of high salt intake increased with the increasing frequency of eating out, which exceeded the recommendation from the World Health Organization of 1.5 g.

Nurses captured that delivery ordering and dining out during meal times was common on a daily basis during work, and they did not have enough time to have a decent and balanced meal with fruits. In addition, sometimes they did not have enough time to prepare meals due to ad hoc working shifts and fatigue. This reduced their motivation to cook or seek for healthy meals, leading to reliance on convenience food with high salt and calories [33]. Furthermore, nurses believed that food sharing could build bridges of communication and enhance established relationships among colleagues. A large and abundant meal was usually prepared for festive events such as Chinese Lunar new year, Christmas, etc. [34]. All of these undoubtedly affected their nutrient intake, or perhaps would exceed the daily requirements of certain nutrients.

### 4.2. Emotional Eating and Overweight/Obesity

Shift work plays a significant role in overweight or obesity in nurses, especially working on night shifts [35]. According to research undertaken by Cheung and Yip in 2015, 53.5% of the participating Hong Kong nurses claimed to have stress symptoms of an emotional burden with increased workload during shift work [36]. The nature of the working units and the position of the nurses was also taken into account [36]. To cope with the stress, nurses chose to eat double or more than usual during or after work [35]. However, they were not necessarily hungry. In this study, such findings are often reported by participants who were obese or overweight. They claimed to have double the portion of their meals after a stressful and busy working shift, and ate whatever they wanted regardless of its nutritional value. Moreover, they worked in the departments which might require a higher concentration of intensive care to the patients (i.e., OBS and ICU). This further increased tension during work and led to emotional eating as a “strategy” to deal with negative emotions [33]. As observed from the FFQ data, participants had excessive total fat and sugar intake, which aligned with their unhealthy dietary habits.

### 4.3. Social Influence as an Enabler or Impediment Towards Healthy Eating

The FFQ data obtained from P1 and P4 were more able to meet the daily dietary requirement in various nutrients, which might contribute to the positive peer and family influence they have experienced, respectively. To supplement the data from P1, the completion of a clinical training program in diabetes allowed her to improve her dietary quality. Although nurses, as professional health care providers, should have basic nutritional knowledge, this enhanced health-related knowledge proved to make this provider more aware of her own healthy food options and diet [5].

This study found that peer influence was regarded as imposing negative impacts on nurses’ behaviors but could also become an enabler towards healthy eating. Individuals tend to adhere to their goals when they are in the group of people with similar aspirations [37]. Another study conducted by Yuan et al. (2018) [38] also claimed that positive peer influence could enable positive attitudes, practices, and behavioral changes for nurses. It could help to improve their eating habits, become less fatigued during work, and ultimately improve their quality of care towards patients [38]. As reported by P1 and P6, their colleagues chose to eat healthily and wisely with no- or low-fat, salt, and sugar diets. They even discussed healthy food options with each other and exercised together out of working hours. This unique ward culture of unity in staying healthy allowed nurses to build up a “self-monitoring” healthy diet and eating style during work, or even at home [39].

This study also found that family influence plays a vital role in shaping an individual’s eating habits and styles; this can affect individuals’ food preferences since childhood which is in line with the literature [40]. The cooking style varies among different families. The healthier the diet that was cooked, the more the risk of obesity and eating disorders would decrease [41]. Due to geographical concerns, especially in Hong Kong, it was more common for the mother-in-law or parents in Asian families than foreign countries to be responsible for cooking in a traditional and balanced diet even if their children had moved out [34]. Nurses could follow the cooking styles from their relatives and prepared a healthy diet of their own [41].

### 4.4. Workplace Support as the Key to Healthy Eating

The long working hours of each shift include the times of at least two meals per day, so it is crucial to have a healthy and balanced diet in the workplace during work [39]. Some of the nurses in this study stated that workplace support could motivate them to eat healthily as a whole. A user-friendly pantry with various cooking utensils allowed nurses to take their own initiative in preparing healthy meals during work. Moreover, nutritional labeling on the food sold from the cafeteria or vending machine was also suggested as a way to allow nurses to choose their diet wisely [39].

Furthermore, the evidence proved that educational programs regarding nutritional knowledge and healthy eating could effectively motivate healthier behaviors during work [42]. By utilizing the enabler of peer influence, programs could be launched together with a reward system for colleagues joining together in eating healthily [39]. Use of meal planning apps or fitness trackers to monitor food intake could also be a new intervention to initiate healthy eating habits during work [43]. All these programs allow nurses to become motivated to maintain a healthy diet. Ensuring adequate and uninterrupted meal and break time is also another key to success [39]. Most importantly, support from senior staff and bosses provided a booster to the nurses to work, as well as to stay healthy.

### 4.5. Strengths and Limitations

This study was the first to fill in the gap in understanding how shift work influences Hong Kong nurses to maintain a healthy and regular diet. Unlike other studies that only understand the barriers and facilitators affecting healthy eating during work through interviews, this study also applied a multi-method study with quantitative data on the dietary patterns of each participant. It could help to obtain a thorough picture on the current health status of nurses as well as the difficulties for them to maintain healthy eating during work. Furthermore, the use of the photovoice methodology could enhance the empowerment of the participants to speak up more with the means of photographs, as well as to prompt dialog between and interviewees and the researcher.

Nevertheless, this study also possessed several limitations. Firstly, the recruitment included RNs only, and not all the ranks of the nurses were involved. Secondly, the findings on the pattern of dietary behaviors, assessed through a validated food frequency questionnaire and a 3-day dietary record, were explorative due to the small sample size, therefore further studies are needed to confirm the results. Thirdly, the working environment of the nurses from different working departments or hospitals might vary, which may not fully reflect the current dietary pattern of nurses in Hong Kong. Lastly, participants might only capture photos with a positive side and those not photographed may be excluded from the discussion, resulting in a possibly biased understanding of the study phenomena [44].

### 4.6. Further Implications of the Study

To minimize the barriers to and promote the facilitators in healthy eating of shift work nurses, it is suggested to enhance peer and workplace support such as the introduction of a monthly fruit day and nutritional guidelines within the hospital as this is observed to be effective in promoting healthy eating culture during work [42]. A reward program could also motivate nurses to stay and eat healthily [42]. This study could act as a blueprint for other health care organizations to be aware of the health conditions of staff, especially on healthy food choices, and therefore to motivate and promote a healthy, balanced diet working environment. Further studies could also involve other health care professionals, such as doctors and physiotherapists other than nurses, to get a broader picture of eating habits and cultures in other health care sectors.

## 5. Conclusions

This study has identified multiple barriers and the facilitators affecting them to maintain healthy eating during work in terms of individual, social, organizational, and environmental factors through a mixed-method of quantitative and qualitative approaches using photovoice. Maintaining a healthy diet for the nurses is not only crucial for their physical and mental well-being, but it also influences their ability to provide high-quality, professional, and safe care to their patients. Health care organizations should therefore foster a healthier workforce by enhancing peer and workplace support such as ensuring adequate meal breaks and developing nutritional guidelines, which could ultimately strengthen both nurse well-being and the quality of care provided to patients.

## Figures and Tables

**Figure 1 nutrients-17-01162-f001:**
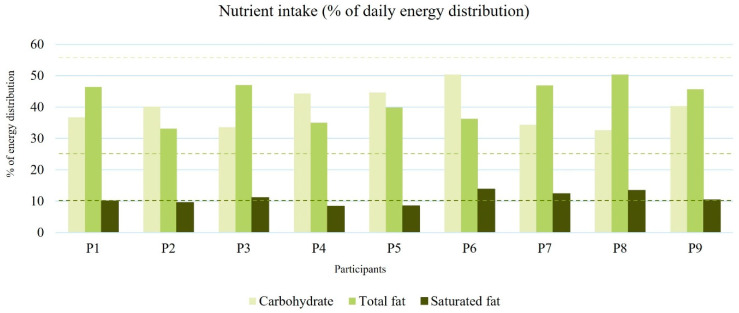
Percentage of daily energy distribution in carbohydrate, total fat, and saturated fat; daily nutrient intake goal for carbohydrate: 55–65%, total fat: 20–30%, saturated fat: <10%. Dashed lines indicate the recommended range for each nutrient based on the DRIs.

**Figure 2 nutrients-17-01162-f002:**
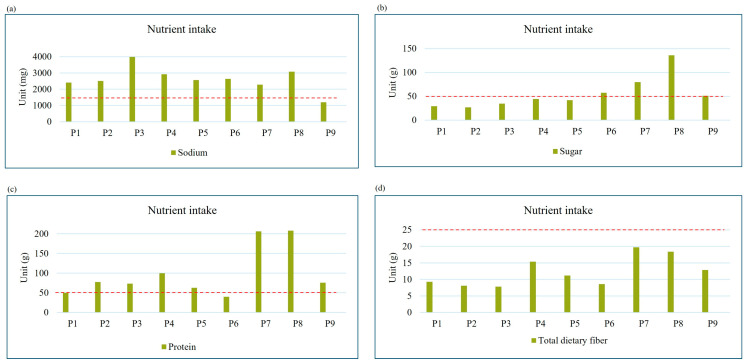
Various nutrient intakes of the participants: (**a**) sodium, (**b**) sugar, (**c**) protein, (**d**) total dietary fiber; daily nutrient intake goal for (**a**) 1500 mg, (**b**) 50 g, (**c**) 50–60 g, (**d**) 25–30 g. Dashed lines indicate the recommended range for each nutrient based on the DRIs.

**Table 1 nutrients-17-01162-t001:** Demographic data.

Participant	Working Department	Sex (Age)	Marital Status	Years of Experience	No. of Night Shift per Month	BMI	BMl Classification *
1	Medical	F (28)	Single (Live with family)	5	5-6	21.9 (160/56)	Normal
2	A&E	F (27)	Single (Live with family)	4	4	18.3 (155/44)	Underweight
3	OBS (midwife)	F (30)	Single (Live with family)	3	5	25.7 (165/70)	Obese
4	Pediatric ICU	F (37)	Married (2 children: 4 and 6 yo)	10	4	21.3 (162/56)	Normal
5	Operating Theater	F (32)	Single (Live with boyfriend)	6	4	20 (155/48)	Normal
6	Surgical	M (42)	Married (No children)	8	5	22.2 (175/68)	Normal
7	Medical	M (50)	Single (Live alone)	25	5	23.9 (177/75)	Overweight
8	ICU	M (29)	Single (Live with family)	6	5	23.8 (169/68)	Overweight
9	Surgical	M (45)	Married (1 child: 15 yo)	18	5	22.2 (18072)	Normal

* Hong Kong Center for Health Protection [18].

**Table 2 nutrients-17-01162-t002:** Categories and sub-categories on the barriers to and facilitators of healthy eating among nurses.

	Barriers	Facilitators
Main Category	Sub-Category	Sub-Category
Individual	Comfort food and emotional eating	Palatability
	Palatability	Personal knowledge
	Personal perception, attitude, and health issues	Perceived healthiness in food choice
Social	Colleagues’ influence/appreciations from patients	Colleagues’ influence
	Family influence	Family influence
Organizational	Shift work nature	Workplace support
	Sudden change of work schedule	
	Meal break availability	
Environmental	Cost and affordability of fast food	Cost and affordability of healthy food
	Availability and accessibility of healthy food during work	Availability and accessibility of healthy food during work

## Data Availability

Data are available upon reasonable request. The data are not publicly available due to ethical and privacy considerations.

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
