# Peer review of "Barriers and Facilitators to Healthy Eating for Shift-Work-Registered Nurses in Hong Kong Public Hospitals: An Exploratory Multi-Method Study"

_nutrients, 2025, doi:10.3390/nu17071162_

Round 1
Reviewer 1 Report
Comments and Suggestions for Authors
Title: Review of the manuscript "Barriers and facilitators to healthy eating for shift work registered nurses in Hong Kong Public Hospitals: an exploratory multi-method study" (nutrients-3515649)
Thank you for the opportunity to review this study, which aims to analyze the eating habits of shift-working nurses in Hong Kong public hospitals. The research involves a sample of nine participants. The topic is highly relevant, and the multi-method approach is a clear strength of the paper.
However, I recommend several revisions and clarifications before publication:
Abstract
Include more information about the sample (e.g., mean age, standard deviation, and gender distribution) to provide a clearer overview of the study.
Keywords
Ensure that the keywords are listed in alphabetical order.
Methods – Recruitment
The authors indicate: “Interested parties were directly contacted the PI through email or phone call.” It would be helpful to clarify exactly how participants were able to express their interest and how the sample was ultimately selected.
Methods – Recruitment Channels
The authors state that “Snowball and purposive sampling were adopted during the recruitment period via social media platforms and emails.” Please specify which social media platforms were used and how the email addresses were obtained.
Measures – Food Frequency Questionnaire (FFQ)
- Calculate and report the internal consistency (e.g., Cronbach’s alpha) of the questionnaire for this sample.
- Provide more information about the FFQ’s response format and include sample items to give readers a clearer understanding of how the instrument was structured.
Results and Discussion
These sections are well-organized and do not raise any particular concerns.
Practical Implications of the Study
Elaborate further on the practical implications of the findings to enhance this section and offer concrete suggestions for healthcare practitioners and the broader scientific community.
Author Response
Reviewer 1
Thank you for the opportunity to review this study, which aims to analyze the eating habits of shift-working nurses in Hong Kong public hospitals. The research involves a sample of nine participants. The topic is highly relevant, and the multi-method approach is a clear strength of the paper.
Response: Thank you for your compliment.
However, I recommend several revisions and clarifications before publication:
Abstract
Include more information about the sample (e.g., mean age, standard deviation, and gender distribution) to provide a clearer overview of the study.
Response:
Added in line 15 as you suggested. Thanks.
Keywords
Ensure that the keywords are listed in alphabetical order.
Response:
Revised as you advised. Thanks.
Methods – Recruitment
The authors indicate: “Interested parties were directly contacted the PI through email or phone call.” It would be helpful to clarify exactly how participants were able to express their interest and how the sample was ultimately selected.
Response:
2.2. Settings and Participants – participants contacted the PI according to the contact information shown on the recruitment poster and the PI selected the participants according to the eligibility criteria.
Methods – Recruitment Channels
The authors state that “Snowball and purposive sampling were adopted during the recruitment period via social media platforms and emails.” Please specify which social media platforms were used and how the email addresses were obtained.
Response:
Thanks for your comments. Social media platforms including Facebook and Instagram have been added in 2.2. Settings and Participants.
Measures – Food Frequency Questionnaire (FFQ)
- Calculate and report the internal consistency (e.g., Cronbach’s alpha) of the questionnaire for this sample.
- Provide more information about the FFQ’s response format and include sample items to give readers a clearer understanding of how the instrument was structured.
Response:
Thanks for your comments.
The FFQ was previously validated in a Hong Kong population, demonstrating good reproducibility (>0.75) and relative validity (r≥0.70) in assessing dietary intake patterns. We have added this in the manuscript (Section 2.3.2).
Sample items and response formats have been added to the supplementary materials (Figure S1).
Results and Discussion
These sections are well-organized and do not raise any particular concerns.
Response: Thank you for the compliment.
Practical Implications of the Study
Elaborate further on the practical implications of the findings to enhance this section and offer concrete suggestions for healthcare practitioners and the broader scientific community.
Response:
Thanks so much for your comments. Implications have been elaborated in 4.6. Further implication of the study.

Reviewer 2 Report
Comments and Suggestions for Authors
The study aims to investigate the dietary habits of shift-working nurses in public hospitals, identifying barriers and facilitators to healthy eating. The authors used a mixed-methods approach.
Introduction is well done. However, please clearly formulate the purpose of the study at the end of the introduction.
Methods and material are well presented. The sample size is too small.
Results. The authors present many barriers and facilitators such as individual, social, organizational, and environmental. They illustrate the findings with statements from the respondents. There is a lot of data against such a small number of respondents.
There is no discussion, only conclusions. The authors wrote: “this study has clearly identified the current dietary habits of the Hong Kong Public Hospitals shift work nurses and the relative barriers and facilitators”. In my opinion, by examining 9 individuals, no conclusions can be drawn about the population of nurses working the Hong Kong Public Hospitals. Such a conclusion is not legitimate. There is no indication of the limitations of the study in the manuscript.
Author Response
Reviewer 2
The study aims to investigate the dietary habits of shift-working nurses in public hospitals, identifying barriers and facilitators to healthy eating. The authors used a mixed-methods approach.
Introduction is well done. However, please clearly formulate the purpose of the study at the end of the introduction.
Response:
Revised the purpose of the study as advised. Thank you.
Methods and material are well presented. The sample size is too small.
Response:
Thanks for your positive comments on methods part.
Regarding the sample size, it aligns with the recommended scope in traditional photovoice methodology. However, there is an increasing trend to use larger sample size in this methodology to enhance the generalizability of findings to a broader population. We have clarified the sample issue within the framework of common photovoice methodology (2.2. Settings and Participants).
Results. The authors present many barriers and facilitators such as individual, social, organizational, and environmental. They illustrate the findings with statements from the respondents. There is a lot of data against such a small number of respondents.
Response: I appreciate your recognition of the richness of qualitative data.
There is no discussion, only conclusions. The authors wrote: “this study has clearly identified the current dietary habits of the Hong Kong Public Hospitals shift work nurses and the relative barriers and facilitators”. In my opinion, by examining 9 individuals, no conclusions can be drawn about the population of nurses working the Hong Kong Public Hospitals. Such a conclusion is not legitimate. There is no indication of the limitations of the study in the manuscript.
Response: Thanks for comments. We’d like to clarify that discussion part spans from line 465-545, while the limitation ranges from line 556-562.
For the conclusions, we agree with you that the findings regarding dietary patterns are not the focus of this study, so we have removed this conclusive sentence. Additionally, we have noted this point in the limitations section due to the sample size.

Round 2
Reviewer 2 Report
Comments and Suggestions for Authors
The authors have improved the manuscript satisfactorily. Everything is fine.